# Determining the relative salience of recognised push variables on health professional decisions to leave the UK National Health Service (NHS) using the method of paired comparisons

Andrew Weyman ,[1] Rachel O'Hara ,[2] Peter Nolan,[3] Richard Glendinning,[1] Deborah Roy,[1] Joanne Coster[2]

¹Department of Psychology, University of Bath, Bath, UK
²School of Health and Related Research, The University of Sheffield, Sheffield, UK
³School of Management, University of Leicester, Leicester, UK

**Correspondence to**
Dr Andrew Weyman;
A.Weyman@bath.ac.uk

## ABSTRACT

**Objective** The primary and secondary impacts from the COVID-19 pandemic are claimed to have had a detrimental impact on health professional retention within the UK National Health Service (NHS). This study set out to identify priorities for intervention by scaling the relative importance of widely cited push (leave) influences.

**Design** During Summer/Autumn 2021, a UK-wide opportunity sample (n=1958) of NHS health professionals completed an online paired-comparisons exercise to determine the relative salience of work-related stress, workload intensity, time pressure, staffing levels, working hours, work–homelife balance, recognition of effort and pay as reasons why health professionals leave NHS employment.

**Setting** The study is believed to be the first large-scale systematic assessment of factors driving staff exits from the NHS since the COVID-19 pandemic.

**Results** All professions gave primacy to work-related stress, workload intensity and staffing levels. Pay was typically located around the midpoint of the respective scales; recognition of effort and working hours were ranked lowest. However, differences were apparent in the rank order and relative weighting of push variables between health professions and care delivery functions. Ambulance paramedics present as an outlier, notably with respect to staffing level (F-stat 4.47, p=0.004) and the primacy of work–homelife balance. Relative to staffing level, other push variables exert a stronger influence on paramedics than nurses or doctors (f 4.29, p=0.006).

**Conclusion** Findings are relevant to future NHS health professional retention intervention strategy. Excepting paramedics/ambulance services, rankings of leave variables across the different health professional families and organisation types exhibit strong alignment at the ordinal level. However, demographic differences in the weightings and rankings, ascribed to push factors by professional family and organisation type, suggests that, in addition to signposting universal (all-staff) priorities for intervention, bespoke solutions for different professions and functions may be needed.

## BACKGROUND

National Health Service (NHS) staff vacancy statistics for 2022 show an increase from approximately 133 100 full-time equivalent

---

### STRENGTHS AND LIMITATIONS OF THIS STUDY

⇒ This is a large-scale systematic assessment of the relative strength of widely cited *push* factors underpinning health professional exit from National Health Service (NHS) employment since the manifestation of the COVID-19 pandemic.

⇒ A key strength of paired comparisons is that the output is an interval scale that provides a quantifiable indication of the distance between the items, in this instance the relative strength of push influences.

⇒ Differences in the rank order and relative weighting of push variables between the principal NHS health profession families are identified.

⇒ Although large for a variable ranking study of this type, the sample size was not sufficient to support multivariate exploration of all permutations of potential interactions between the health profession demographics and push variables of interest.

---

staff in the quarter to June 2022 to around 133 400 in the quarter to September 2022. This represents a 5-year high (data are not available prior to the quarter from April to June 2017). The overall vacancy rate in the quarter to September 2022 stood at 9.7%, which also represents a 5-year high.[1] There are widespread claims, aligned with more substantiative evidence that the experience of working through the pandemic has significantly diminished health professional resolve/capacity to remain in NHS employment.[2–4] NHS staff survey data (2020)[5] shows a 44% increase in the proportion of staff reporting work-related stress, and around a fifth reporting considering exiting NHS employment, both of which have been attributed to the pandemic experience.[6] There are also reports of significant numbers of staff, notably junior doctors and nursing assistants, leaving to take-up better paid jobs

offering better working conditions outside the NHS.[6 7] It is important to note that the disparity between staff expressions of intention to quit and leaving behaviour can reflect systematic bias, potentially exaggerating the potential number of leavers. However, as a source of common error across successive waves of the NHS staff, trend data can be considered more reliable.

The pent-up demand for care alone has created a major imbalance between demand for treatment and the capacity to deliver.[8] Recognition of the finite scope for recruitment of migrant labour and the inevitable time lag in training new health professionals, combined with institutional worry over the likely magnitude of pandemic-associated exits, has given rise to an unprecedented focus of NHS policy makers on finding ways to stabilise/enhance staff retention. This paper reports findings from a study of the relative salience and weighting of an array of widely cited *push* influences on health professional exit from NHS employment.

From the perspective of intervention aimed at enhancing retention rates, it is important to know which variables exert the strongest push and their relative weightings. However, it is also important to determine whether this profile varies across different segments of the workforce, that is, whether 'one-size-fits all' or bespoke interventions are needed.[9] Almost all previous studies of NHS staff retention are unable to answer this question, as their samples are typically limited to single professions or institutions.[10]

The study is the first large-scale systematic analysis of the precursors to exit following the emergence of COVID-19. The findings are relevant to human resource intervention strategy aimed at stabilising and enhancing staff retention. They provide evidence of which variables constitute the greatest influence on employee leave decisions and the degree to which they present as universal (all staff) or variable across different health profession and organisation-type demographics. The focus here is on health professional migrations to non-NHS employment, as distinct from transitions from one NHS employer to another, the former representing a loss to state sector capacity, the latter producing no net change. This key distinction is frequently overlooked or blurred in reports of NHS staff turnover. The research builds on the research team's previous work within the sector[10] and broader published findings on reasons why health professionals leave NHS employment.

## Established insights

Prepandemic systems perspective research focused on the role of extrinsic aspects of work in eroding job satisfaction, mental health, physical health and strength of attachment to the NHS, notably job demands, workload, pay, resources and configuration of working hours.[11–13] The perspective, in essence, is a risk-based attrition model focused on the detection of precursor relationships. A high proportion of study findings point to shortage of resources, particularly staff resources as a, if not *the*, cornerstone element harbouring the potential to produce an array of detrimental ripple effects. Staff shortages tend to produce time pressure, increased workload and work rate, norming of (paid and unpaid) overtime working, with implications for employee fatigue, work–homelife balance and sickness–absence rates, mental health and burnout.[14–17] Related psychosocial effects identified include corrosive effects on morale, frustration and concern over compromised standards of patient care,[18 19] amplified worry over capacity to cope, making errors and, relatedly, anxiety over personal professional vulnerability.[20–23]

At a fundamental level, issues of high/excessive workload can be conceptualised as reflecting an imbalance between the demand for patient care relative to resource, although other climate and culture factors can play a role, for example, claims of increased performance monitoring, other forms of auditing and accountability-related bureaucracy have also been reported as notable sources of health professional frustration and disaffection.[24 25] Contributions from systems perspectives on mental health and work-related stress are also extensive and principally emphasise the role of contextual effects arising from the design and configuration of work, that is, extrinsic elements of job demands, that embody the potential to cause psychological harms that erode staff disposition/capacity to remain.[26 27]

Findings on the role of pay in exit decisions are mixed. Some claim it as a primary criterion,[28 29] whereas others report it as less important than workplace climate variables, notably sufficiency of resources and staffing levels.[13 30 31] Pay is relevant in both absolute and relative terms, with linkages to notions of fairness, equity and recognition of effort.[32–35] As Dean[36] notes, dissatisfaction with pay tends to be amplified where rates remain static and are perceived by employees to be disproportionate to effort expended in the presence of significant rises in work rate and workload, this being the case since the emergence of COVID-19.

The issue of working hours, with respect to shift duration, flexibility over their configuration and the availability of part-time work, has received extensive attention.[37] Findings relating to the widespread adoption of compressed hours and associated increases in typical shift duration (typically 10–12 hours) over recent decades within the NHS are mixed, liked by some staff while disliked by others.[38–40] Irrespective of employee preferences, there are claims of long-term detrimental impacts on health and amplified fatigue among older workers.[37] Increasing the availability of flexible and part-time hours is widely cited as a means of enhancing retention.[39 41–43]

The analysis that follows measures NHS health professionals' ratings of the relative salience of headline push influences on exit decisions after 12 months of exposure to COVID-19 working conditions.

The aim of the study was to provide future human resource policy-relevant insight into issues and priorities

for intervention to sustain/enhance NHS staff retention rates.

The operational objectives were (1) to produce on ordinal high–low ranking of headline push variables; (2) to produce scaled output of the relative weighting of headline push variables; and (3) to determine the degree to which objectives 1 and 2 vary by health profession family and type of secondary care organisation (acute care; mental health, community and ambulance).

## METHODS
### Patient and public involvement
There was no patient or public involvement in this study as its focus was on NHS employees only. Participants in the study have access to a report on headline findings from the UK-wide survey of NHS staff.

### Data collection
The data were gathered from July to December 2021 as a component of the second wave of the authors' UK-wide online quantitative survey of influences on NHS staff retention.[44 45]

Mirroring previous research into NHS staff retention by the authors,[10] we selected paired comparisons as the method for the study.[46] Paired comparisons is well suited to eliciting views on multifaceted subjective variables. The cognitive load on participants is low (each is requested to select one item from a randomly presented pair of items and to repeat this for all permutations of pairings within the set). Its key strength relative to alternatives, for example, direct ranking, subjective rating scales and sorting techniques such as Q.Sort and repertory grid, is that the output is an interval scale that provides a quantifiable indication of the distance between the items,[46–48] in this instance the relative salience of push influences.

### Generation of the item set
We selected a set of eight widely referenced push variables,[16 49 50] six of which were common to a previous study of NHS staff retention[10]: *staffing levels, working hours, mental health/stress, pay, time pressure* and *recognition of contribution. Workload intensity* and *work–life balance* were added to the item set following consultation with government, employer, professional body and trade union stakeholders, and to take account of the prevailing post-pandemic working conditions, workload intensity and work–life balance were added to the item set.

### Reference criterion
The study set out to tap health professionals' insight into and perceptions of the relative importance of a set of widely cited reasons why colleagues within their health profession/job role leave NHS employment. Specifically, participants were asked 'how important the following issues are to explain why (their health profession inserted) staff *leave the NHS*'.

**Table 1** Breakdown of sample by job–family

| Profession job–family | Employed, n (%) (NHS England only) | Paired ranking sample, n (%) |
|---|---|---|
| Doctors (consultant & specialist, including trainees)[62] | 124 000 (15) | 227 (12) |
| Nurse professionals and midwives[62] | 332 000 (40) | 687 (35) |
| Nurse, non-professional[62] | 280 000 (33) | 384 (20) |
| Allied health[62 63] | 84 000 (10) | 417 (21) |
| Ambulance[62] | 18 000 (2) | 243 (12) |
| Total | 837 000 | 1958 |

This produced a set of subjective scales of push effects. The purpose was not to determine rates of exit but to compare how staff beliefs about these push factors might vary between different health professional families and types of care provider organisation. The underpinning rational here was twofold. First, asking staff about their own future leave versus stay intentions was not considered a robust option as it tends to exaggerate leaving rates to an unknowable degree due to recognised intention behaviour disparities[51]; that is, people often do not achieve their intentions. Asking participants about the behaviour of colleagues cannot be considered free from bias or error as it is subjective, but the error can be considered common across the different groups of personnel. Second, from the perspective of identifying human resource priorities for intervention aimed at stabilising/improving retention rates, it is important to know the relative strength of the push factors in the item set as these can be considered to operate as precursors to exit.

### Participants
Participants were a UK-wide volunteer sample recruited via the YouGov panel, UNISON trade union members and 12 NHS trusts in England using electronic mailing and newsletter communications to distribute a link to the online survey. A breakdown of the sample by health profession job–family is provided in table 1.

### Procedure
Participants were presented with pairs of push factors, for all permutations of the eight factors, 28 for each participant. The order of presentation of pairs was randomised. For each pair, participants were asked, 'Which of these two factors is the bigger influence on why staff in your profession/job role leave the NHS' (see online supplemental appendix 1).

### Preanalysis check of within-respondent consistency
To consider the data set suitable for scale development, it is important to determine that participants are able to make consistent judgements; that is, evidence that the stimulus items produced logically inconsistent output, of the type A>B>C>A, would suggest that the item set does

not scale and would be unsuitable for scale development. Therefore, tests of within-respondent consistency (Kendall's $K$) were performed on a pilot sample of response sets (n=60). This revealed that 94% of response sets exhibited high consistency ($K \geq 0.70$). The item set therefore was judged to be suitable for reliable scale development.[46 47]

## Analysis

The analysis took the form of an iterative complementary approach.

► Generation of an NHS-wide all-profession family interval scale to determine the global profile of relative salience of the push variables in the item set (see online supplemental appendix 1).

► Generation of separate, dedicated, health profession family and care delivery organisation-type scales to determine the degree of homogeneity/heterogeneity in the rank orders and weightings of the set of push variables between these demographics.

► Generation of push variable scales relative to pay and staffing levels, by health profession family and care delivery organisation type.

► Formal (statistical) testing of the degree of variability in the collective strength of push variables relative to pay and staffing level across different demographics; that is, do variables within the domain of reasons to leave constitute a stronger push with some professions/organisation types than others?

## Global (all NHS health professions) ranking of push variables

In the first instance, a global all-NHS (all profession/all delivery function) interval scale of the relative importance of push influences was produced. Judgement proportions were determined and means for each push were calculated. Setting the lowest-ranked variable (recognition of contribution) to 0 and the highest (mental health/stress) to 100 produced the relative weightings depicted in online supplemental appendix 2.

Findings point to the primacy of mental health/stress and staffing levels, pay is ranked fourth of the eight push variables, and recognition of contribution and working hours occupy the two lowest ranks. When interpreting the findings, it is important to keep in mind that they relate to a scaling of widely cited headline drivers of exit, that is, a scaling of *important push* influences. As such, a low ranking is not synonymous with being unimportant.

Having established the global profile, deeper analyses explored differences by health profession and type of care provider organisation.

## Health profession (job–family) contrasts

Contrasts were explored between doctors (consultants and specialists, including trainees), nurse professionals and midwives, nurse non-professionals, allied health professionals and ambulance paramedics (including technicians; see table 1) (scientific and technical professionals were excluded due to restricted sample size) and four types of care provision setting: acute, mental health, community and ambulance services.

This analysis revealed a notable alignment between profession groupings but also difference in rank orders and respective weightings. Mirroring the global profile, mental health/stress, staffing levels and workload intensity dominated the top three ranks across the different professions. However, staffing level was ascribed a notably less prominent position by ambulance participants (see online supplemental appendix 3). Formal testing (analysis of variance (ANOVA)) revealed that that contrast with the other job families was statistically significant (F-stat 4.74, p value 0.004).

Working hours, recognition of effort and time pressure consistently occupied the three lowest ranks across all profession families, except ambulance services, where working hours was ranked fifth and ascribed a markedly higher weighting. Pay consistently occupied the mid-range, being ranked fourth or fifth across each of the occupation groupings, although having a higher weighting among nursing (most notably among nursing support grades, job bands 1–4) and ambulance service paramedics. The most marked contrast between the profession families related to the relative weighting of work–home life balance. Again, ambulance service paramedics present as an outlier, ranking this the second strongest influence on exit with a notably higher weighting (83). There was high consistency with respect to the ranking (fourth or fifth) of this variable across the other professions, although allied health gave it a higher weighting (48) than doctors (31.6), nurse/midwife professionals (35.5) and nursing support (28.7).

Correlation (Pearson $r$) and linear regression ($R^2$) analyses, comparing variable ratings for all permutations of pairings of job–family, show high alignment across all groups ($r$=0.93–0.99, $R^2$=0.88–0.97) with the exception of paramedics, which present as having a different profile ($r$=0.57–0.72, $R^2$=0.44–0.53). An alternative and complementary means of comparing the degree of between-group agreement is to compare the degree of alignment at the ordinal level. Formal testing of this (kappa coefficient $K$) highlighted high congruence between nursing staff (professional and non-professional) and doctors ($K$=0.75), but modest to low alignment between other groups (nurse professional vs allied health, doctors vs nurse non-professional, $K$=0.50; nurse professional vs allied health vs doctors, $K$=0.375; all other contrast, $K$=0.25). Confirming the regression analysis, we found the most marked contrast between paramedics and each of the other occupations.

## Service provider organisation contrasts

The findings indicated high alignment by type of care provider organisation at the level of rank order, excepting ambulance services, which again present as an outlier. However, a number of contrasts are apparent with respect to the weighting, notably *work intensity* in acute care hospitals and community, time pressure in community and

**Table 2** Normalised assessment values calculated from paired comparisons made by five NHS health profession families

| Push variable | Nurse professional job band 5+ | Nurse non-professional job band 4– | Allied health | Paramedics | Doctors |
|---|---|---|---|---|---|
| Time pressure | −0.331 | −0.328 | −0.197 | −0.193 | −0.402 |
| Working hours | −0.421 | −0.413 | −0.356 | 0.0004 | −0.521 |
| Work intensity | −0.081 | −0.096 | 0.020 | 0.143 | −0.159 |
| Work–home balance | −0.257 | −0.274 | −0.151 | 0.330 | −0.400 |
| Pay | −0.197 | −0.120 | −0.166 | 0.117 | −0.328 |
| Mental health | 0.0055 | 0.069 | 0.070 | 0.485 | −0.084 |
| Recognition | −0.425 | −0.402 | −0.342 | −0.253 | −0.589 |
| Staffing level | 0.000 | 0.000 | 0.000 | 0.000 | 0.000 |

Graphical representations of these findings depicting the relative distance between push variables are provided in online supplemental appendix 5.

*recognition of effort* in mental health settings (see online supplemental appendix 4).

Correlation analyses of variable ratings for all permutations of pairings of care provider organisation show a contrast between ambulance service and the other functions (acute vs mental health, acute vs community, mental health vs community; $r \geq 0.9$, $R^2 > 0.90$). Ambulance services versus acute, community and mental health; $r = 0.50–0.62$, $R^2$ 0.30–0.38). The other three functions show strong alignment over the five highest-ranked variables, with mental health and community sharing a common order of priority.

### Weightings of push variables relative to staffing resource and pay

Reflecting Thurstone's five methods,[46] the analysis proceeded to scaling the push variables relative to knowable objective value anchors of staffing levels and pay. For these separate analyses, the respective anchor item (pay or staffing) was set to zero. This produced a push scale for each profession and type of care provider organisation. Formal testing of between group differences (ANOVA) afforded insight into whether the set of headline push influences relative to pay (and staffing level) exerts

stronger collective leverage on some professions and care provider organisation types than others.

### Staffing levels

Table 2 gives the relative weightings by job-family. It highlights paramedics/ambulance services as an outlier. The profession contrasts revealed a statistically significant difference between paramedics and doctors and paramedics and qualified nurses (F-stat 4.29, p value 0.006) but did not exceed the 0.05 threshold in comparison to allied health and nursing support.

Contrasts by type of care provider (table 3) showed a significant difference between ambulance and the other care provider functions, acute (F-stat 3.98, p value 0.018), community (p value 0.040) and mental health (p value 0.056).

### Pay

In the case of pay, contrasts were explored in relation to profession and (where the sample size was sufficient) by grade, in the case of nurses (professionals vs nursing support). Table 4 gives the relative weightings by job–family.

**Table 3** Normalised assessment values calculated from paired comparisons made by four NHS healthcare provider organisation types

| Push variable | Mental health | Acute | Community | Ambulance |
|---|---|---|---|---|
| Time pressure | −0.283 | −0.323 | −0.228 | −0.193 |
| Working hours | −0.396 | −0.412 | −0.411 | 0.0004 |
| Work intensity | −0.090 | −0.078 | −0.037 | 0.143 |
| Work–home balance | −0.208 | −0.248 | −0.225 | 0.330 |
| Pay | −0.162 | −0.188 | −0.193 | 0.117 |
| Mental health | 0.066 | −0.001 | 0.029 | 0.485 |
| Recognition | −0.342 | −0.413 | −0.402 | −0.253 |
| Staffing level | 0.000 | 0.000 | 0.000 | 0.000 |

Graphical representations of these findings depicting the relative distance between push variables are provided in online supplemental appendix 6.

**Table 4** Normalised assessment values calculated from paired comparisons made by five NHS health profession families

| Push variable | Nurse professional job band 5+ | Nurse non-profssion job band 4– | Allied health | Paramedics | Doctors |
|---|---|---|---|---|---|
| Time pressure | −0.134 | −0.208 | −0.031 | −0.309 | −0.074 |
| Staffing level | 0.197 | 0.119 | 0.166 | 0.116 | 0.328 |
| Working hours | −0,244 | −0.292 | −0.191 | −0.116 | −0.193 |
| Work intensity | 0.116 | 0.024 | 0.185 | 0.026 | 0.169 |
| Work–home balance | −0.060 | −0.154 | 0.014 | 0.213 | −0.071 |
| Mental health | 0.203 | 0.189 | 0.235 | 0.369 | 0.244 |
| Recognition | −0.228 | −0.283 | −0.176 | −0.369 | −0.261 |
| Pay | 0.000 | 0.000 | 0.000 | 0.000 | 0.000 |

Formal testing (ANOVA) revealed no statistically significant differences (F-stat 0.35, p value 0.842). A graphical representation of the relative distance between pay and the other push variables by job–family is provided in figure 1.

## DISCUSSION

Determining the relative strength of headline factors driving early exit offers the promise of informing strategic decision making with respect to ameliorative intervention both by government and NHS employers. Insights in this area are of central relevance to signposting which push effects constitute priority issues for intervention with the potential to deliver the highest impact on health professional retention and future NHS capacity. The overarching finding is that mental health/stress, staff resources and workload show primacy in the scaling of push influences. Pay and work–homelife balance occur within the mid-range of the distribution, and recognition of effort and working hours occupy the lowest positions.

Beyond almost universal consensus over the primacy of mental health/stress and the basal position ascribed to working hours and recognition of effort, a number of health profession and organisation-type demographic contrasts are apparent in the rankings and relative weightings ascribed to push influences. The most marked contrast is between the paramedic/ambulance services profile and other occupations/functions. Staffing was ranked sixth by paramedics, compared with first or second, and work–homelife balance was ranked second,

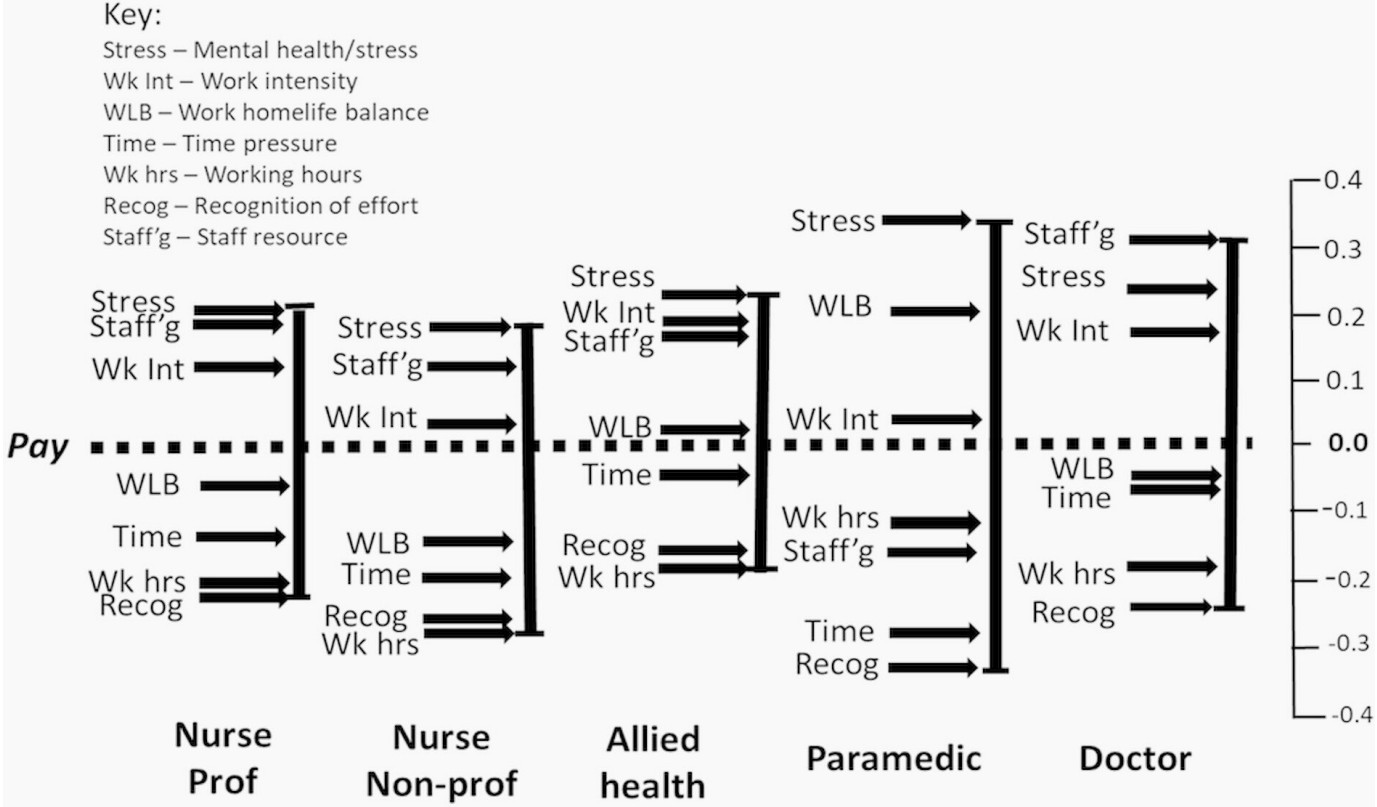

**Figure 1** Weightings of *push* variables referenced to pay, by profession.

compared with fourth or fifth by each of the other occupations. The weighting ascribed to staffing by the other professions indicates that it exerts a stronger push on doctors, allied health, and nurse professionals than non-professional nurses, and paramedics. The higher salience of work–life balance to paramedics may reflect differences in the (ir)regularity and duration of their working hours. Emergency ambulance crews have markedly more irregular shift patterns, combined with routinised overtime working. A significant proportion of their overtime is attributable to shift 'over-runs', that is, ad hoc, unplanned and involuntary extensions to the working day/incursions into non-work time.[52] The higher weighting of working hours by paramedics compared with the other professions aligns with this conclusion.

The much lower rank ascribed to staffing resource by paramedics is perhaps the most noteworthy finding, in view of its primacy in the scales produced by each of the other personnel groups/functions, as well as the strength of objective evidence of significant shortage of paramedics.[53] It is possible that the relative social isolation of paramedics has the effect of making the issue of staff shortages less cognitively available, that is, less visible and immediately apparent in their work environment, than would be the case in a hospital context. Accepting this, a similar profile might have been expected from community care respondents, a high proportion of whom are also subject to peripatetic lone working, but this was not apparent. On balance, it seems questionable that social isolation would be sufficient to significantly attenuate employee awareness of staff shortages, not least because of its visible secondary impacts, that is, increased pressure to work overtime, threats to the realisation of trust call-out targets, as well as regular high-profile coverage of staff shortages within ambulance sector publications and mass media.[54–56]

Relative to staffing, stress, work–home life balance, work intensity, pay and working hours exert a stronger push influence among paramedics than the other professions; that is, within the leave domain, these variables exert a relatively stronger push for paramedics than other professions. This explanation rests on the premise that staff shortages can be treated as constant across each of the professions and care delivery functions, a shared benchmark against which other push variables can be scaled. If this interpretation is accepted, it suggests that attention to paramedic retention represents a key priority and source of vulnerability to the NHS. The finding from the research team's broader survey of NHS staff retention that 24% of paramedics (n=1157)[44] reported having submitted one or more applications for a non-NHS job in the previous 6 months, compared with the all-staff rate of 13% (n=9220),[45] lends weight to this conclusion. The data do not permit a precise explanation for the marked difference in the paramedic response profile. However, the evidence that the profile of precursors to exit for paramedics is different from the other professions and functions presents as strong.

In contrast to some contemporary media and industrial relations accounts,[55–57] and some academic research findings,[58] pay, as a reason to leave did not feature prominently. Ranked fourth/fifth by each of the professions, this profile mirrored finding from previous research on NHS staff retention,[10] which concluded that dissatisfaction with pay as an incentive to leave needs to be considered with reference to opportunities for more remunerative employment elsewhere. UK Labour Force Survey data indicate that (excepting doctors and dentists and care assistants) pay rates achieved by NHS leavers are not significantly higher, and for certain professions (notably, paramedics), they are commonly lower.[6 37 53] While other variables appear to exert a stronger push than pay, this is not grounds to diminish it as a potential source of dissatisfaction in absolute terms. However, it does suggest that attention to pay alone is unlikely to fix the retention issue.

The finding that working hours was ascribed a low rank and weighting across all professional groups and functions (again excepting paramedics) mirrors its profile in previous work on staff retention[16] and questions the emphasis it receives within contemporary policy and employer guidance publications.[42 43] Widely cited as attractive to all employees, but particularly millennials and staff with caring responsibilities, increasing the availability of part-time and flexible hours is cast as key to increasing retention rates. However, the findings suggest that its salience to leave decisions is lower than mental health, staff resource (excepting paramedics), work intensity, pay and work–homelife balance. Mirroring the conclusions on pay, this does not represent grounds to diminish the value of flexible work hours and its attractiveness to staff, but it does suggest that attention to this issue may have a modest impact in the absence of attention to more highly ranked variables.

The findings point to the primacy of mental health/stress as the reason why staff leave. Mental health and stress are lag effects that can be attributable to work and non-work variables, as well as intersects between the two, mediated by individual resilience. For employers, a comprehensive perspective on intervention means addressing root causes attributable to the configuration of work, as well as supporting individuals at times of need.[59 60] In contrast to stress, sufficiency of staff resources is an important lead (as well as a lag) indicator. Sufficiency of resources has material impacts on job demands, work–life balance, time pressure and working hours, as well as psychosocial impacts on psychological stress and morale.

### Strength and limitations

This study is believed to be the first large-scale systematic assessment of push variables following manifestation of the COVID-19 pandemic and its secondary impacts. The findings provide insight into the relative weighting of headline reasons why health professionals leave NHS employment. The output is relevant to determining human resource priorities for intervention to stabilise and enhance staff retention rates.

The method of paired comparisons produces an interval scale of the relative distance, within psychophysical space, between a set of related entities, which can also be anchored to respondents' subjective estimates of knowable quantifiable entities, for example, pay or staffing level. Respondents were asked about their beliefs regarding the exit behaviour of peers rather than their own intentions. Neither feature can be considered free from some degree unquantifiable error. However, each can be regarded as a source of common error, that is, that *cancels out*, when comparing the output profiles across different segments of the NHS workforce, in relative terms.

The findings reported within this paper are based on an opportunity sample of voluntary participants with notable variability in the sizes of the occupation and care organisation-type subsamples. In common with the NHS annual staff survey, all other voluntary participation employee surveys, the potential for self-selection response bias cannot be discounted.

Proportionately, the profession job–family subsamples show alignment with NHS contemporary staff-in post ratios for doctors, qualified nurses and midwives, but under-represent nurse non-professionals and over-represent allied health professionals and ambulance staff. However, the smallest subsample exceeded accepted norms regarding the minimum number of participants for scale development[61] by a factor of 10. Also, it is important to note than the scales of push factors are not subject to variability due to the sample size or influenced by variation between the proportions in realised sample and actual proportions within the NHS workforce.

Although large for a variable ranking study of this type, the sample was not sufficient to support multivariate exploration of possible interactions between the health profession and organisation-type demographics of interest.

## CONCLUSIONS

Excepting paramedics, rankings of leave variables across the different health professional families exhibit a high degree of alignment, at the ordinal level, and highlight the primacy of psychological stress, staff shortages and work intensity. However, the presence of demographic differences in order of priority and weightings of push variables suggests that, in addition to signposting universal priorities for intervention aimed at stabilising and stemming prevailing exit rates, it is important to consider bespoke priorities/solutions for different professions and functions. Paramedics present as an outlier, exhibiting a profile that is significantly different from other health professionals. There may also be grounds for inferring that the leave variables explored exert a relatively stronger push on members of this group.

While increases in pay are transparently important to NHS staff, findings from this research suggest that enhancements in that domain alone may produce a modest impact on retention. An equivalent conclusion might be drawn with respect to the current high-profile emphasis on increased access to flexible working hours as a solution within contemporary NHS staff retention guidance to employers. Both have potential to *do good*, but there are grounds for inferring there is a risk that neither may deliver sufficient *good* to redress the high and rising exodus in the absence of attention to what present as more fundamental factors driving exit. Importantly, scope for addressing the highest-ranked factors driving exit, in large degree, lies beyond the gift of NHS employers.

The findings of this paper relate to the scaling of widely cited headline factors driving exit, that is, a scaling of important push influences. As such, it would be unwise to interpret low-ranked variables as unimportant. A more prudent interpretation would be to regard intervention to address low-ranked variables as embodying potential to contribute to a comprehensive, multifaceted programme of activity, but unlikely to produce the necessary degree of leverage in the absence of attention to more highly ranked, more fundamental, influences of staff exit behaviour.

## Recommendations

► The insights from this study can contribute to the evidence base for prioritising push issues and demographics for intervention aimed at stabilising/enhancing NHS health professional retention.

► NHS workface policy and planning functions could review the extent to which current staff retention strategies align with factors identified in this study.

► Further research/published evidence synthesis is needed to map the scope for intervention to address the higher ranked push factors.

► The set of push factors can be considered to constitute precursors to exit. NHS employers would benefit from a tool to benchmark and regularly (e,g, annually) monitor these variables in order to gain feedback on the effectiveness of their interventions and practices aimed at stabilising/enhancing health professional retention rates.

**Acknowledgements** The authors thank the interview participants and are also grateful for the input of the project advisory group for their guidance and assistance in facilitating access to the participants. The authors also thank members of the research stakeholder advisory group involving representatives from NHS England/Improvement, health sector professional associations and trades unions for their guidance and feedback on the content of the UK-wide survey of NHS staff, of which the paired ranking exercise was a component.

**Contributors** AW, PN, RO'H and RG conceived and designed the study. AW led the data analysis and drafted the paper. DR and JC managed the process of securing necessary ethics and governance permissions to conduct the study. DR contributed to the data analysis. All authors read drafts of the manuscript and approved the final version. AW is guarantor for the paper.

**Funding** Economic and Social Research Council, No: ES/V015389/1.

**Competing interests** None declared.

**Patient and public involvement** Patients and/or the public were not involved in the design, conduct, reporting or dissemination plans of this research.

**Patient consent for publication** Not applicable.

**Ethics approval** This study involves human participants and was approved by the University of Bath, Department of Psychology Ethics Committee (reference number: PREC 20–259) and the NHS Ethics Committee. The research received National Institute for Health research portfolio adoption. Respondent participation was voluntary and anonymous, and all participants provided informed consent prior to taking part.

**Provenance and peer review** Not commissioned; externally peer reviewed.

**Data availability statement** Data are available upon reasonable request. The data set can be made available to researchers.

**ORCID iDs**
Andrew Weyman http://orcid.org/0000-0001-6292-3394
Rachel O'Hara http://orcid.org/0000-0003-4074-6854

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
