## [Reviewer comments · BMJ Open]

ARTICLE DETAILS

TITLE (PROVISIONAL)	Determining the relative salience of recognised push variables on Health Professional decisions to leave the UK National Health Service (NHS), using the method of paired-comparisons.
AUTHORS	Weyman, Andrew; O'Hara, Rachel; Nolan, Peter; Glendinning, Richard; Roy, Deborah; Coster, Joanne

VERSION 1 – REVIEW

REVIEWER	Moscelli, Giuseppe University of Surrey, School of Economics
REVIEW RETURNED	16-Feb-2023

GENERAL COMMENTS	I have read your manuscript with interest, and I suggest you the following edits to improve its quality, readability and impact: • You should change the title, it is misleading, as you only use participants from the English NHS, not the whole UK.• Abstract: please specify that “Setting: This is believed to be the first study that has attempted to produce a large-scale systematic assessment of drivers of exit” within the NHS.• Abstract: This sentence “During summer/autumn 2021, a UK-wide opportunity sample (N=2028) of NHS health professionals completed an on-line paired-comparisons exercise.” should go into Design, not Setting. Also, it is not UK-wide but England-wide.• Abstract. Rephrasing “push influences” and “push drivers” with “push factors”, in abstract and throughout the whole text.• Abstract and manuscript. The use of the term “demographic” with respect to the push variables does not make sense. These are “work-related factors” or “working conditions”. Please replace within the whole manuscript, it is really misleading as “demographic” is generally used for staff characteristics such as age, gender, nationality, ethnicity, etc...• Missing information. Please provide, in an Appendix or Online Appendix, an example of the paired-comparisons questionnaires that were given to the study participants.• Missing information. Please also clarify how the following factors were defined to the study participants, before they performed the paired comparisons: work intensity; staffing levels (related to the whole Trust? Related to the participant’s work category within the Trust? Or how?); work-life balance. If these terms/concepts were not properly defined to participants prior to their interviews, this must be made explicit.• Missing information. Please clarify the complete process of recruitment and interview of the participants. This info is missing, but crucial, as the timing (i.e. months, year) of the interviews matters with respect to the push factors implied by the COVID-19 pressures
--

	on healthcare services.  • Methods. Please clarify that the responses are respondents' opinion with respect to exit decisions of colleagues. This is unclear and misleading, it comes too late in the paper (page 22, line 6). • Methods. Please report statistics about the characteristics of the participants in the pilot sample used to check the item scaling. • Statistics. The notation is very imprecise. Please replace f- with F-stat, and p with p-value. • Statistics. Please provide descriptive statistics about the characteristics of the study participants (e.g. mean age, gender, nationality, ethnicity, push factors) by provider type or professional role, and the same descriptive statistics for these characteristics at the population level in the NHS, e.g. using NSS data. It is important to assess how representative this selected sample is with respect to the NHS workforce population at the time of the study (or the closest period). This output must be included in the main body of the paper. • Contents. Figure 1, the point "Generation of an NHS-wide all-profession family interval scale to demine the global profile of relative salience of the push variables in the item set." and the paragraph "Global (all NHS health professions) ranking of push variables" should go to an Appendix, as they do not add any value to the discussion and the important insights of the paper, which rely on the heterogeneity of the rankings of the push factors for different NHS professions and healthcare provider types. • Contents. Page 6, line 59. Which are the "rates" here? Unclear. • Contents. Please remove all "our", e.g. "common to our previous study of NHS staff retention", page 9 line 11, with an impersonal alternative, e.g. "common to a previous study of NHS staff retention" • Contents. Page 13, line 10. Questionable use of the term "discrete", better to be replaced with "different". Similarly at page 20, line 5. • Contents. Tables 5 and 6 can go to an Appendix, if their contents is the same as Figures 4 and Figure 6. • Contents. Page 17, line 34, it should be "[Insert figure 6 about here]", not figure 5. • Contents. The language used in the Discussion section can be simplified and streamlined, e.g. rephrasing wording such as "an array of demographic contrasts", "At the level of rank order,", "Reference to the weighting ascribed to staffing" • Contents. Please clarify/expand on the definition of "lead effect" for staff resources, page 21, line 29. It is unclear, and questionable, why staff resources should not be considered as both a lag and lead effect. • Contents. In the study Limitations, page 21, please include a broader discussion about the use of an opportunity sample in this study, in particular if some of the characteristics of the sample participants are different from the mean of the NHS workforce population, to be reported as requested above.
--	--

REVIEWER	Garside, Joanne University of Huddersfield
REVIEW RETURNED	17-Feb-2023

GENERAL COMMENTS	Reviewer feedback: An interesting paper which is contemporary and of relevance to the intended audience. Overall, the paper is written well and presents transparent data and robust methods.
--

	The background section provides an overview of the knowledge on attrition in the health professions. The research context is presented in the Post Covid-19 world – the rationale for focusing on Post C19 needs further discussion or removing completely as this automatically dates the research rather than providing the contemporary context. In addition, vacancy figures are highlighted across 2022 1st and 2nd Quarter therefore not providing a broad context to the study. The authors provide NSS data that 1/5th of staff is considering leaving – further discussion about the translation to actual leavers is needed. The generation of push variables are clear although unless mental health/stress is counted as 2, I count 5 not 6 variables as suggested on p9. It is not until p22 that we understood that the participants were asked about their 'beliefs regarding the exit behaviours of peers' rather than their own intentions. This changes the context and needs discussion and clarification earlier in the paper. Intention to leave and actual leavers is a key challenge for researchers in this field so please explain/defend this approach fully. Please include clear priority recommendations for policy and further research. Minor suggestions: Abstract: p2 - Opens with a very long sentence; p3 spelling error; p10 point i. spelling error Keywords: should include attrition, retention Please put nurses and midwives (midwives will be offended to appear in brackets!) Overall I really enjoyed reading this paper and look forward to seeing it in print.
--	---

VERSION 1 – AUTHOR RESPONSE

Reviewer 1

Dear authors,

I have read your manuscript with interest, and I suggest you the following edits to improve its quality, readability and impact:

4. You should change the title, it is misleading, as you only use participants from the English NHS, not the whole UK.

The sample is not limited to England, it is all UK . The text has been amended to emphasise this in the description of the sample on P.7 of the revised draft.

5. Abstract: please specify that “Setting: This is believed to be the first study that has attempted to produce a large-scale systematic assessment of drivers of exit” within the NHS.

Text revised to read: ‘This is believed to be the first study that has attempted to produce a large-scale systematic assessment of drivers of exit within the NHS since the emergence of the COVID-19 pandemic. ‘

6. **Abstract:** This sentence “During summer/autumn 2021, a UK-wide opportunity sample (N=2028) of NHS health professionals completed an on-line paired-comparisons exercise.” should go into Design, not Setting. Also, it is not UK-wide but England-wide.

This sentence has been migrated to design the section of the abstract.

7. **Abstract.** Rephrasing “push influences” and “push drivers” with “push factors”, in abstract and throughout the whole text.

The text has been amended to factors driving – throughout the manuscript

8. **Abstract and manuscript.** The use of the term “demographic” with respect to the push variables does not make sense. These are “work-related factors” or “working conditions”. Please replace within the whole manuscript, it is really misleading as “demographic” is generally used for staff characteristics such as age, gender, nationality, ethnicity, etc... The term demographic(s) is used throughout the text in its conventional sense to relate to statical character sits of population segments, a central focus of the being to compare and contrast the ranking and weighting of push variables between different health profession and care provider organisation types. The text has been amended to make this more transparent in the abstract; p5, para 2; p18, p2; p23 para 1; p22 para 2.

9. **Missing information.** Please provide, in an Appendix or Online Appendix, an example of the paired-comparisons questionnaires that were given to the study participants.

A copy of the instructions presented to participants for completing the paired comparisons task, a list of the push variables, the list of permutations of pairings and an explanatory paragraph have been added at appendix 1.

10. **Missing information.** Please also clarify how the following factors were defined to the study participants, before they performed the paired comparisons: work intensity; staffing levels (related to the whole Trust? Related to the participant’s work category within the Trust? Or how?); work-life balance. If these terms/concepts were not properly defined to participants prior to their interviews, this must be made explicit.

The participants were not interviewed –a statement on page 7; the first sentence of the Method section in the original manuscript reads ‘The data was gathered from July to December 2021, as a component of the second wave of our UK-wide survey of influences on

NHS staff retention’. The words ‘on-line questionnaire/quantitative survey’ have been added to the revised draft to emphasise the format in which the data was gathered . Respondents were asked to rate the push factors with reference to staff in your profession / job role. An additional sub-section on ‘Procedure’ has been added to the Method section to clarify the question asked of participants and the to clarify the reference object In addition, the reader is referred to appendix 1 for a copy of the format used.

11. **Missing information.** Please clarify the complete process of recruitment and interview of the participants. This info is missing, but crucial, as the timing (i.e. months, year) of the interviews matters with respect to the push factors implied by the COVID-19 pressures on healthcare services.

As outlined noted above (response 7), the participants were not interviewed. Text has been added to section (iii) Participants that 'recruitment was conducted using electronic mailing and newsletter communications to distribute a link to the online survey'.

12. **Methods.** Please clarify that the responses are respondents' opinion with respect to exit decisions of colleagues. This is unclear and misleading, it comes too late in the paper (page 22, line 6).

This issue is addressed within revised Method section in the added sub-sections) (ii) Reference criterion (iv) Procedure.

13. **Methods.** Please report statistics about the characteristics of the participants in the pilot sample used to check the item scaling.

It would be possible to determine the occupational demographics of the pilot sample. However, that would be potentially misleading – as the purpose of piloting in paired comparisons relates to the properties of the stimulus material (in this case the push item set). Hence, it is not a pilot in the sense of a representative sample of health professionals – but a test of the signal intensity of the stimulus material, i.e. a test of cognition with respect to stimulus and response – which would be considered universal rather than a source employee demographic variability. An analogy would be the capacity to reliably and consistently distinguish between the relative intensity of a set of different light sources, or the shades of a set of colours. The underpinning psychophysics proofs and assumptions for the method paired comparisons can be found in references 46 and 47 (cited on p.10) for readers with an interest in the method. We could add text to explain the above – but our view would be that this would be disproportionate for a widely used method (the method is for example specifically referred to in a recent NIHR invitation to tender on NHS staff retention). Also by analogy, this degree of explanation would not routinely be provided for other widely used methods, for example, when claiming that the assumptions for a factor analysis had been met.

14. **Statistics.** The notation is very imprecise. Please replace f- with F-stat, and p with p-value. Test amended to F-stat and p-value.

15. **Statistics.** Please provide descriptive statistics about the characteristics of the study participants (e.g. mean age, gender, nationality, ethnicity, push factors) by provider type or professional role, and the same descriptive statistics for these characteristics at the population level in the NHS, e.g. using NSS data. It is important to assess how representative this selected sample is with respect the NHS workforce population at the time of the study (or the closest period). This output must be included in the main body of the paper.

There are a number of obstacles to determining accurate staff numbers for the principal health profession families / groupings in the NHS.

Reviewer 1 suggested the NHS staff survey as potential source. Unfortunately, as participating in the NHS staff survey is also a voluntary, so in common with our sample is subject to similar respondent self-selection bias.

Numbers quoted in NHS and other publications on staff are prone to vary between sources. NHS England Staff-in-post offers the more comprehensive and consistent source, excepting

allied health professionals (there are also reasonable grounds for considering these to generalise to the all-UK profile). Relatedly, while it might be possible to derive data at deeper levels of granularity, for each profession job-family by age, ethnicity, gender etc, there is a risk that this implies greater accuracy than could be robustly substantiated. Different sources on NHS staff numbers also exhibit variability for common reference periods, and while the error may be common over time within each, this feature reduces confidence in determining numbers in absolute terms. Significant reporting inaccuracy within NHS Trust/Board submissions is also acknowledged in conversations we have had with NHS statisticians.

Table 1 has been revised to include a breakdown of proportions of health professions employed in the NHS in spring 2022, to provide alignment with the period in which our ranking data was collected.

Text has also been added to the acknowledgement of limitations (p. 24 of the revised draft) to clarify that the scales of push factors for each job-family are not subject to variability due to the sample size for each profession or influenced by variation between the proportions in our sample and actual proportions within the NHS workforce.

16. Contents. Figure 1, the point “Generation of an NHS-wide all-profession family interval scale to demine the global profile of relative salience of the push variables in the item set.” and the paragraph “Global (all NHS health professions) ranking of push variables” should go to an

Appendix, as they do not add any value to the discussion and the important insights of the paper, which rely on the heterogeneity of the rankings of the push factors for different NHS professions and healthcare provider types.

Figure 1 has been moved to appendix 2, and signposted within the revised text. Additionally, in response to a request from BMJ editorial office to reduce the total number of figures and tables, figures 2,3,4, & 5 from the initial draft of the paper have been moved to appendix 3, 4, 5 & 6, respectively.

17. Contents. Page 6, line 59. Which are the “rates” here? Unclear.
Text amended within the revised draft ‘rates; deleted – (p.6 line 9)

18. Contents. Please remove all “our”, e.g. “common to our previous study of NHS staff retention”, page 9 line 11, with an impersonal alternative, e.g. “common to a previous study of NHS staff retention” .

The text of the revised draft has been amended: p8 line 4;p8 line 11; p.8 line 21; p17 line 11; p19 line 21; p.19 line 22; p.2- line 7; p.20 line 14; page 20 line 19; page 21 line 1; page 21 line 17; page 24 line 3.

19. Contents. Page 13, line 10. Questionable use of the term “discrete”, better to be replaced with “different”. Similarly at page 20, line 5.
The text amended within the revised draft to ‘different’ p.20 lines 2 & 7

20. Contents. Tables 5 and 6 can go to an Appendix, if their contents is the same as Figures 4 and Figure 6.

The content of these figures is different and conveys insights on demographic contrasts that are of relevance to NHS HR intervention focus. We therefore believe that they should remain in the main body.

21. Contents. Page 17, line 34, it should be “[Insert figure 6 about here]”, not figure 5. The text of the revised draft has been amended to ‘insert figure 6 about here’

22. Contents. The language used in the Discussion section can be simplified and streamlined, e.g. rephrasing wording such as “an array of demographic contrasts”, “At the level of rank order,”, “Reference to the weighting ascribed to staffing”

A number of amendments have been made to simplify and streamline the text within the discussion

23. Contents. Please clarify/expand on the definition of “lead effect” for staff resources, page 21, line 29. It is unclear, and questionable, why staff resources should not be considered as both a lag and lead effect.

The text has been amended with the revised draft on page 21 lines 9 & 10

24. Contents. In the study Limitations, page 21, please include a broader discussion about the use of an opportunity sample in this study, in particular if some of the characteristics of the sample participants are different from the mean of the NHS workforce population, to be reported as requested above.

Additional text has been added to that relates to the participant sample within the section on limitations, on page 24 of the revised draft.

Reviewer: 2

An interesting paper which is contemporary and of relevance to the intended audience. Overall, the paper is written well and presents transparent data and robust methods.

The background section provides an overview of the knowledge on attrition in the health professions.

25. The research context is presented in the Post Covid-19 world – the rationale for focusing on Post C19 needs further discussion or removing completely as this automatically dates the research rather than providing the contemporary context. In addition, vacancy figures are highlighted across 2022 1st and 2nd Quarter therefore not providing a broad context to the study.

The revised introductory paragraph has been added at the beginning of the background section (page4). However, we believe that the post-COVID-19 context should be retained. Our rationale here is that our research funding from UKRC was to explore (primary and longer-term secondary) impacts arising from the COVID-19 pandemic and its legacy. While we accept that the push variables we explored are not pandemic specific, to our knowledge there has been no

systemic assessment of their profile, in particular their relative salience since the manifestation of COVID-19 in early 2020. Moreover, there are strong grounds for believing that the post-COVID period has witnessed an intensification of a number of headline push factors.

26. The authors provide NSS data that 1/5th of staff is considering leaving – further discussion about the translation to actual leavers is needed.

A paragraph acknowledging and explaining the limitations of intention evidence has been added to p4 para 1.

27. The generation of push variables are clear although unless mental health/stress is counted as 2, I count 5 not 6 variables as suggested on p9.

Time pressure was omitted in error - now added to the revised draft on p.9, line 9.

28. It is not until p22 that we understood that the participants were asked about their 'beliefs regarding the exit behaviours of peers' rather than their own intentions. This changes the context and needs discussion and clarification earlier in the paper. Intention to leave and actual leavers is a key challenge for researchers in this field so please explain/defend this approach fully.

A section has been added under the heading (ii) Reference criterion (pages p9 and 10 of the revised draft)

29. Please include clear priority recommendations for policy and further research. A section on Recommendations has been added (pages 24 and 25) of the revised draft.

Minor suggestions:

30. Abstract: p2 - Opens with a very long sentence; p3 spelling error; p10 point i. spelling error
The sentence has been divided into two shorter sentences

31. Keywords: should include attrition, retention
Attrition has been added and retention was already included within the key words.

32. Please put nurses and midwives (midwives will be offended to appear in brackets!)

Brackets have been removed to read 'nurse professionals and midwives'.(p12 penultimate paragraph) and Table 1

Overall I really enjoyed reading this paper and look forward to seeing it in print.

VERSION 2 – REVIEW

REVIEWER	Moscelli, Giuseppe University of Surrey, School of Economics
REVIEW RETURNED	11-Jun-2023

GENERAL COMMENTS	I am satisfied with the edits made and the revised manuscript,
--

	although both the punctuation and phrasing of some of the text could be sensibly and easily improved. Some relevant examples: 1) "Objective: This study, located within the context of widespread claims that primary and secondary impacts from the COVID-19 pandemic have had a detrimental impact on health professional retention. It set out to provide staff-retention policy-relevant insight into priorities for intervention to sustain/enhance health professional (doctor, nurse, allied health and ambulance paramedic job families) retention rates by determining the relative salience of widely cited push influences on exit from employment in the UK National Health Service (NHS)." This paragraph's first sentence seems to be without a verb and it is also the first time I see that the objective is "this study". The authors can do a better job. An alternative: "The primary and secondary impacts from the COVID-19 pandemic are claimed to have had a detrimental impact on health professional retention. This study is set out to [...]" 2) "summer/autumn 2021." Missing capital letters for Summer and Autumn. 3) "NHS staff vacancy statistics for 2022, show an increase [...]" should be without the comma.
--	---

REVIEWER	Garside, Joanne University of Huddersfield
REVIEW RETURNED	31-May-2023

GENERAL COMMENTS	Thank you for addressing my comments - I have attached some very minor edits - contact publisher to view file.
--

VERSION 2 – AUTHOR RESPONSE

Comments received from reviewers

1 - The objective .has been recast to: Objective: The primary and secondary impacts from the COVID-19 pandemic are claimed to have had a detrimental impact on health professional retention within the UK National Health Service (NHS). This study set out to identify priorities for intervention by scaling the relative importance of widely cited push (leave) influences.

2 - summer and autumn have been capitalised to: Summer and Autumn - within the abstract.

3 - The comma has been removed from the Sentence 'NHS staff vacancy statistics for 2022... ' (Background: Page 1, Para 1).